# Brain and Muscle Oxygen Saturation Combined with Kidney Injury Biomarkers Predict Cardiac Surgery Related Acute Kidney Injury

**DOI:** 10.3390/diagnostics11091591

**Published:** 2021-08-31

**Authors:** Wiktor Szymanowicz, Ludmiła Daniłowicz-Szymanowicz, Wojtek Karolak, Maciej Michał Kowalik, Romuald Lango

**Affiliations:** 1Department of Cardiac Anaesthesiology, University Clinical Center, 80-210 Gdańsk, Poland; 2Department of Cardiology and Electrotherapy, Medical University of Gdańsk, 80-210 Gdańsk, Poland; ludmila.danilowicz-szymanowicz@gumed.edu.pl; 3Department of Cardiac and Vascular Surgery, Medical University of Gdańsk, 80-210 Gdańsk, Poland; wkarolak@gmail.com; 4Department of Cardiac Anaesthesiology, Medical University of Gdańsk, 80-210 Gdańsk, Poland; mkowalik@gumed.edu.pl (M.M.K.); romuald.lango@gumed.edu.pl (R.L.)

**Keywords:** acute kidney injury, biomarkers, NIRS, tissue oximetry

## Abstract

Background: Early identification of patients at risk for cardiac surgery-associated acute kidney injury (CS-AKI) based on novel biomarkers and tissue oxygen saturation might enable intervention to reduce kidney injury. Aims: The study aimed to ascertain whether brain and muscle oxygenation measured by near-infrared spectroscopy (NIRS), in addition to cystatin C and NGAL concentrations, could help with CS-AKI prediction. Methods: This is a single-centre prospective observational study on adult patients undergoing cardiac surgery using cardiopulmonary bypass (CPB). Brain and muscle NIRS were recorded during surgery. Cystatin C was measured on the first postoperative day, while NGAL directly before and 3 h after surgery. Results: CS-AKI was diagnosed in 18 (16%) of 114 patients. NIRS values recorded 20 min after CPB (with cut-off value ≤ 54.5% for muscle and ≤ 62.5% for the brain) were revealed to be the most accurate predictors of CS-AKI. Preoperative NGAL ≥ 91.5 ng/mL, postoperative NGAL ≥ 140.5 ng/mL, and postoperative cystatin C ≥ 1.23 mg/L were identified as independent and significant CS-AKI predictors. Conclusions: Brain and muscle oxygen saturation 20 min after CPB could be considered early parameters possibly related to CS-AKI risk, especially in patients with increased cystatin C and NGAL levels.

## 1. Introduction

Cardiac surgery-associated acute kidney injury (CS-AKI) is a well-recognized, but still incompletely understood, clinical problem that significantly impacts short- and long-term outcomes [1,2]. Early identification of patients at risk for this complication may allow clinicians to introduce interventions to decrease the risk of CS-AKI [3]. Besides oliguria, serum creatinine level and its changes are the parameters that serve as the criteria for CS-AKI diagnosis; however, it is not an ideal parameter during short-term changes in kidney function because it lags behind the decline and recovery of glomerular filtration rate (GFR) by days [4]. Therefore, there is a need for more sensitive and specific markers for CS-AKI prediction and early recognition.

Data from the literature reveals the possible role of numerous novel acute kidney injury biomarkers. Among them, serum cystatin C and urinary and serum neutrophil gelatinase-associated lipocalin (NGAL) belong to sensitive and clinically useful biomarkers in AKI detection [5,6,7,8,9,10,11,12,13]. Additionally, as lower oxygen delivery during and after cardiopulmonary bypass (CPB) is a significant risk factor of CS-AKI [14], the measurement of tissue oxygenation by near-infrared spectroscopy (NIRS)—a modern, non-invasive technique, seems of great clinical value [14,15,16,17,18,19]. Previous publications on NIRS monitoring concerning brain protection during extensive aortic surgery [20] or direct renal oxygenation measurement in infants have confirmed its importance [21,22,23]. The role of tissue oximetry monitoring by the NIRS technique in adults in various cardiac surgery procedures has been extensively studied, but the results are inconsistent [15,19,24,25]. Our study aimed to evaluate if regional cerebral oxygen saturation (rScO_2_) and somatic oxygen saturation of thenar muscles (SomO_2_), in addition to blood NGAL and cystatin C concentration, could allow for better CS-AKI prediction in adult patients undergoing cardiac surgery with the use of a cardiopulmonary bypass.

## 2. Materials and Methods

### 2.1. Study Design and Patient Selection

The study protocol conformed to the Ethical Principles for Medical Research Involving Human Subjects outlined in the Declaration of Helsinki was approved by the local institutional review board for scientific studies (NKBBN/122/2014) and registered in the Clinical Trials database (NCT02979275). Between December 2016 and November 2018, all adult patients (≥18 years old) undergoing scheduled cardiac surgery using CPB were prospectively included in this pilot study. Patients with a left ventricular ejection fraction of less than 30% and/or estimated glomerular filtration rate (eGFR) < 30 mL/min/1.73 m^2^ were excluded from the study. Of 125 patients enrolled in the study, 11 were excluded due to lack of consent. Written informed consent was obtained from all participants.

CS-AKI was the primary outcome of this study. Patients were divided into two groups: those who developed CS-AKI (AKI group) and those without CS-AKI (non-AKI group). CS-AKI was diagnosed according to the Kidney Disease Improving Global Outcomes (KDIGO) classification system, which defines CS-AKI [26] based on their serum concentration of creatinine: CS-AKI was defined as an increase in creatinine level by 0.3 mg/dL over 48 h or 50% in 7 days; creatinine level the day before the operation was used as the baseline. Three stages of kidney damage were taken into account: stage 1 (1.5 to 1.9-fold increase in the initial creatinine concentration or increase above 0.3 mg/dL), stage 2 (2 to 2.9-fold increase in the initial creatinine concentration), and stage 3 (3-fold increase in baseline creatinine concentration or increase in creatinine concentration to the value ≥ 4 mg/dL or the need for renal replacement therapy). Anaesthetic management and postoperative sedation in the studied patients are presented in Text S1, whereas surgical and cardiopulmonary bypass management is described in Text S2.

### 2.2. NIRS Monitoring

Tissue saturation was continuously monitored using an INVOS monitor (INVOS-TM 5100C Cerebral Somatic Oximeter, Covidien, Mansfield, MA, USA); the sensors were placed on the thenar muscle of the right hand—opposite to the radial arterial catheter (SomO_2_) and left side of the forehead (rScO_2_). The absolute NIRS value instead of its relative change was registered. NIRS data were reported at discrete time points instead of the AUC of NIRS.

### 2.3. Laboratory Tests

Blood NGAL was assessed by point-of-care testing (Triage Meter NGAL Test, Biosite, Alere Health, San Diego, CA, USA) via a rapid fluorescence immunoassay. The 2 mL blood sample was aspirated from the arterial line in an EDTA-anticoagulated syringe, and several drops of blood were added immediately on the sample port of the Alere Triage^®^ Meter device, which provided the resulting printout within 15 min.

The 5 mL arterial blood samples for cystatin C assessment were centrifuged directly after sampling at 4000 rpm in 10 min, at 4 °C, and deep-frozen immediately at −70 °C until laboratory analysis. The concentration of cystatin C in heparinized plasma was determined using molecularly amplified immunonephelometry using Siemens BN II/BN Pro Spec systems. Serum creatinine was measured using the enzymatic method (Abbott Diagnostics Inc., Santa Clara, CA, USA). Blood gas analysis, haemoglobin, and lactate concentration were assessed with ABL800 Flex 835 blood analyser (Radiometer, Copenhagen, Denmark).

Blood NGAL concentration was measured in the operating room directly before surgery and 3 h after the operation in every patient. Blood samples for cystatin C concentration were collected on the first postoperative day (12 to 20 h after the surgery). Serum creatinine concentration was assessed on admission to the hospital, in the vast majority—one day before surgery, as well as 24 and 48 h after the surgery. SomO_2_, rScO_2_, and haemoglobin concentration, were recorded in the following nine time-points of operation: before anaesthesia induction {1}, directly before skin incision {2}, after sternum opening {3}, 20 min after aortic cross-clamping {4}, 40 min after aortic cross-clamping {5}, 20 min after aortic cross-clamp removal {6}, 20 min after weaning from CPB {7}, 40 min after weaning from CPB {8}, and 60 min after weaning from CPB {9}. All the parameters mentioned above were compared between the AKI and non-AKI groups.

### 2.4. Statistical Analysis

Continuous variables were presented as median, quartiles, and range, while categorical data were presented as proportions. Data were tested for normal distribution with the Shapiro–Wilk test. Comparisons between AKI and non-AKI groups were performed with the Mann–Whitney *U* test for continuous variables and Pearson’s chi-square test for categorical variables. Kruskal–Wallis ANOVA test was performed to assess inter-group differences over time. Regarding the parameters checked during surgery, differences for groups and time were presented as figures only for parameters that differed significantly between groups. The accuracy of measured parameters as potential CS-AKI predictors was determined based on the area under the receiver-operating characteristic curve (AUC ROC). The cut-off values with AUC equal to 0.7 and higher were accepted for further calculations.

Additionally, sensitivity, specificity, and positive and negative predictive values (PPV, NPV) were calculated. Logistic regression analyses were performed to determine which parameters with pre-specified cut-off values had the most decisive influence on CS-AKI occurrence. The Pearson linear correlation test assessed the linear dependence of individual variables. *p*-values less than 0.05 were considered significant. Statistical analysis was performed with STATISTICA 12.0 (StatSoft, Tulsa, OK, USA) and R 2.15.2 software.

## 3. Results

Baseline demographic characteristics, clinical and laboratory parameters, and data regarding surgery are presented in Table 1. Among 114 enrolled patients, 18 (16%) met CS-AKI based on the KDIGO criteria, of which 12 were stage 1, in 4—stage 2, and in 2—stage 3 (one of these patients, apart from the diagnostic increase in creatinine level oliguria, was diagnosed in the postoperative period—this patient required renal replacement therapy in the postoperative period). AKI patients were significantly older and demonstrated a slightly higher EURO Score; however, the difference of the latter did not reach statistical significance. Before surgery, median serum creatinine concentration was significantly higher, and haemoglobin concentration was lower in patients from the AKI group. CS-AKI was observed less frequently in patients undergoing aortic valve surgery without concomitant procedures. Aortic cross-clamp time was significantly longer in patients in the AKI group. Patients in the AKI group were noted to have a longer time to extubation, required higher doses of catecholamines on the second postoperative day, were reported to have reduced urine output, and were more likely to require furosemide on the first postoperative day, and this cumulated to a less negative fluid balance on the first postoperative day. AKI group was characterized by higher C-reactive protein (CRP) concentration on the first postoperative day, and higher white blood cell (WBC) count on the third day after surgery. All measured kidney injury biomarkers: blood NGAL before surgery, blood NGAL after surgery, and postoperative cystatin C were significantly higher in the AKI group. The Kruskal–Wallis ANOVA test performed to assess inter-group differences over time demonstrated that haemoglobin concentration, SomO_2_, rScO_2_ were markedly lower in AKI patients (Figure 1, Figure 2 and Figure 3). There were no significant differences in any other measured parameters.

In ROC analyses for NIRS, we identified that rScO_2_ and SomO_2_ measured 20 min after CPB had cut-off values with acceptable AUC levels around 70% (Table 2), while NIRS measured at other time points had lower accuracy for CS-AKI prediction. For blood NGAL (both preoperative and postoperative) and cystatin C concentrations, established cut-off values had a sufficient AUC level and were characterized by high PPV (Table 2).

Plasma cystatin C measured between 12 and 20 h after surgery, at the level ≥1.23 mg/L, was recognized as the most accurate predictor of CS-AKI with AUC 91% (95% confidence interval [CI] 82.0–100.0) (Figure 4).

In univariate logistic regression analysis, the aforementioned cut-off values for blood NGAL, cystatin C, rScO_2_, and SomO_2_ were identified as significant CS-AKI predictors (Table 3).

As a total of 18 AKI were documented among 114 patients, the maximum number of predictors that could be used in a multivariate model without the risk of its over-fitting was 2. Therefore, we tested combinations of rScO_2_ and SomO_2_ 20′ min after CPB with haemoglobin level as an essential factor of lower oxygenation in bivariate logistic regression analysis. rScO_2_ and SomO_2_ measured 20′ after CPB turned out to significantly predict CS-AKI, independently from haemoglobin level measured at the same time: OR 0.15 (0.04–0.59), *p* < 0.007 for SomO_2_ and 0.10 (0.02–0.52), *p* < 0.006 for rScO_2_. Additionally, the combined analysis of biomarkers and NIRS parameters allows for better CS-AKI prediction, revealing the highest power for a combination of cystatin C with rScO_2_ and SomO_2_ (Table 3). We revealed that the correlations between biomarkers and NIRS parameters were weak; although some were statistically significant, the linear correlation coefficient r was below 0.5 (Appendix A). In the additional logistic regression analyses we checked the associations of pre- and intraoperative variables which differed between AKI and non-AKI groups (Table 1) using pre-specified in ROC analyses cut-off values, and association with AKI were as follows: age ≥ 71–OR 3.29 (1.16–9.32), *p* = 0.031, EURO Score ≥ 4.5–OR 6.27 (1.36–28.89), *p* = 0.008, serum creatinine before surgery ≥ 0.84–OR 10.66 (1.36–83.5), *p* = 0.006, haemoglobin concentration before surgery < 13.5–OR 4.27 (1.3–17.32), *p* = 0.019, CPB time ≥ 139 min–OR 3.61 (1.17–11.14), *p* = 0.036, aortic cross-clamp time ≥ 84 min–OR 3.98 (1.19–13.25), *p* = 0.028, three valves surgery–OR 5.47 (1.73–17.32), *p* = 0.006. Additionally, we adjusted NGAL, cystatin C, and NIRS (20′ after CPB) for all these parameters, confirming the significant independence of biomarkers and NIRS in predicting the incidence of CS-AKI (Appendix A).

Trying to find any other predictors of CS-AKI, we additionally compared the patients with lower and higher than 45% LVEF, revealing that the patients with LVEF 30–45% have worse NGAL levels before and after the surgery and worse NIRS parameters (particularly SomO_2_) at some time points (Appendix A). The rate of AKI did not differ between the groups; however, more advanced statistical analyses were not possible due to the small sample size.

## 4. Discussion

The main findings of the present study are that brain and muscle oximetry monitoring based on the NIRS technique and biomarkers (blood NGAL and cystatin C) could be promising tools in predicting kidney injury after cardiac surgery with the use of CPB. We showed that muscle and brain NIRS monitoring, in addition to increased blood NGAL and cystatin C levels, could help to predict CS-AKI. Amongst all time-points taken into account during surgery for NIRS measurement, 20 min after weaning from CPB was revealed to be the most crucial for CS-AKI prediction.

The comparison of the groups in terms of clinical data showed a significant difference in age and a statistically borderline difference in risk calculated using the EuroSCORE scale, confirming the previous literature data [3,27,28]. Excluding patients whose higher risk would be associated with significant left ventricular systolic dysfunction and significant renal function deterioration could be responsible for the relatively low incidence of CS-AKI in the studied patients. However, data from the literature show that CS-AKI may also occur in patients with normal left ventricular systolic function and undisturbed kidney function before surgery [29,30]. Our results are in line with these data.

In the presented study, patients who developed CS-AKI were characterized by a higher creatinine concentration before surgery. The observation that elevated creatinine levels before surgery may be associated with a higher risk of CS-AKI confirms the reports of previous authors [3,27,28] and could be explained by the significantly reduced renal functional reserve in patients with higher creatinine levels and, thus, increased sensitivity to damaging factors by kidneys having fewer active nephrons.

The results of this study also confirm the data from the literature on the influence of haemoglobin concentration on the occurrence of kidney damage after surgery, which was demonstrated, among other things, in a multicentre study involving 3500 patients from seven academic centres [31]. In the presented study, in patients who developed CS-AKI, a lower haemoglobin concentration was found; however, the mean values were within the normal range, which suggests the need for careful assessment of these parameters in patients prepared for cardiac surgery with the use of CPB, and consideration of additional preparation of patients.

In the present study, it was noted that the differences between the AKI and non-AKI groups in terms of haemoglobin and creatinine levels before surgery may suggest that, especially in patients with lower haemoglobin and higher creatinine levels before surgery with KPU, extending monitoring by tissue oximetry, as well as undertaking interventions aimed at increasing tissue saturation could contribute to reducing the risk of CS-AKI, but resolving this issue would require further research.

In the presented study, several additional features distinguishing patients who developed CS-AKI were observed. This complication was prevalent in patients undergoing surgery on two valves (mitral and aortic) and three heart valves. Much less often, CS-AKI was reported in patients who underwent only aortic valve surgery, which may be due to the specificity of patients undergoing this procedure, most often including normal left ventricular function before surgery, less frequent hemodynamic disturbances in the postoperative period, and a shorter time of aortic clamp insertion, compared to multiple valve surgery and mitral valve surgery [32]. The observation that patients who developed CS-AKI required a longer duration of ventilator therapy and that elevated parameters of the inflammatory response (CRP and white blood cell count) could be associated with a higher risk of infection is consistent with the literature data [28,29].

Statistically, both the significantly extended time of applying the transverse clamp to the aorta and the insignificantly longer duration of CPB in the AKI group can be interpreted as a factor contributing to its development, both by activating the inflammatory reaction stimulated by extracorporeal circulation [33] and by more prolonged, persistent disorders of organ perfusion, including renal function, as have been observed during extracorporeal circulation [34,35,36]. The study group was probably too small to show a statistically significant influence of other known factors contributing to the development of CS-AKI related to the operating procedure, which may include, among other things, CPB time [37].

The role of cystatin C and NGAL in CS-AKI prediction has been demonstrated in several papers [5,6,7,8,9,10,11,12,13]. In one of the recent studies [12] on patients undergoing elective cardiac surgery, Wang et al. confirmed that increased serum cystatin C level, with cut-off values pre-specified at different time-points, was related to an enhanced risk of CS-AKI. In our study, based on a similar group, the cut-off value for cystatin C was in line with the cited research. Increased NGAL levels allowed us to detect subclinical kidney injury, even in the absence of a diagnostic increase in serum creatinine, typical for AKI [6]. Precise diagnostic accuracy for NGAL in early prediction of CS-AKI in adults with normal baseline renal function has been reported [9]. In our study, we confirmed this finding on patients without significantly altered kidney function before the surgery. The majority of data from literature concerned NGAL measurement after the surgery, while pre-operative NGAL could also help predict CS-AKI [8,10], as was confirmed in our study.

Until now, the usefulness of brain NIRS was described in a few studies on cardiac surgical patients [15,24,38,39]. Thenar muscle oximetry was also assessed in various clinical settings, including cardiac surgery [40,41]. The authors suspected that low SomO_2_ values might better correlate to acute kidney injury because kidney and muscle vasculature is more sensitive to vasoconstrictors than the brain. To the best of our knowledge, this is the first study where specific NIRS cut-off values of rScO_2_ and SomO_2_ were calculated. In our research, amongst different time points, NIRS measured 20 min after weaning from CPB with a cut-off value of 54.5% for SomO_2_ and 62.5% for rScO_2_, presented the best predictive accuracy for CS-AKI. Current views on brain and muscle saturation are presumptive, and clinically verified algorithms to recover decreased NIRS parameters are lacking [19,42]. The NIRS data reported at discrete time points, instead of the NIRS AUC, were analysed to enable its correlation with arterial blood gas and haemoglobin concentration assessed at the same time-points.

There is no evidence to suggest only a long-lasting, or even short decrease in NIRS levels after CPB, is efficient for CS-AKI development. However, it seems plausible that immediate real-time intervention aimed at the normalization of rScO_2_ and SomO_2_ could be helpful in CS-AKI prevention. On the other hand, the low sensitivity of these parameters is a real weak point of these measures, therefore, the addition of NIRS parameters to cystatin C or blood NGAL level calculation could be a promising direction in the selection of at-risk patients. It would be essential to check the usefulness of NIRS measurements according to the AKI severity; unfortunately, due to limited numbers of AKI patients in every stage, the statistical power of possible analyses would be weak. These issues however need to be confirmed in further investigations on larger groups of patients

## 5. Study Limitations

Our study presents some limitations. Firstly, the group was small, and the number of CS-AKI incidences was low, which meant that we were unable to perform an additional analysis regarding the AKI stage. Similarly, we were also unable to perform a complex multivariate analysis as the maximum number of predictors that could be used in a multivariate model without the risk of its over-fitting was only 2. Other studies measured similar parameters in larger groups of patients, but all analysed biomarkers and NIRS parameters in kidney injury prediction separately. In contrast, our study confirmed the usefulness of the combined measurement of these indices. A further limitation was that we did not measure rScO_2_ and SomO_2_ after surgery, which may well be very important. Next, we used the creatinine level the day before the operation as the baseline; however, we did not have precise information concerning preoperative creatinine levels within the three months before the surgery, which could exclude AKI before the surgery. Additionally, we did not collect the clinical data concerning the excluded patients, therefore, it is difficult to compare these patients with the enrolled population in light of AKI occurrence. Similarly, it is difficult to explain the low percentage of angiotensin-converting enzyme inhibitors and sartans before the study; the ultimate decision regarding pharmacological treatment before the operation in any patient was left to the discretion of the treating physician. We also did not perform a further follow-up; therefore, we cannot discuss the influence of the measured parameters in the long-term, and especially the long-term renal function of the studied patients.

## 6. Conclusions

Brain and muscle oximetry monitoring based on the NIRS technique with particular attention to their values 20 min after weaning from CPB could be considered as early parameters possibly helpful for the increased risk of CS-AKI, especially in patients with increased cystatin C and blood NGAL levels. Our pilot study needs to be extended to a larger group of patients with more events. Otherwise, whether interventions aiming to increase rScO_2_ and SomO_2_ values can decrease CS-AKI risk requires further studies.

## Figures and Tables

**Figure 1 diagnostics-11-01591-f001:**
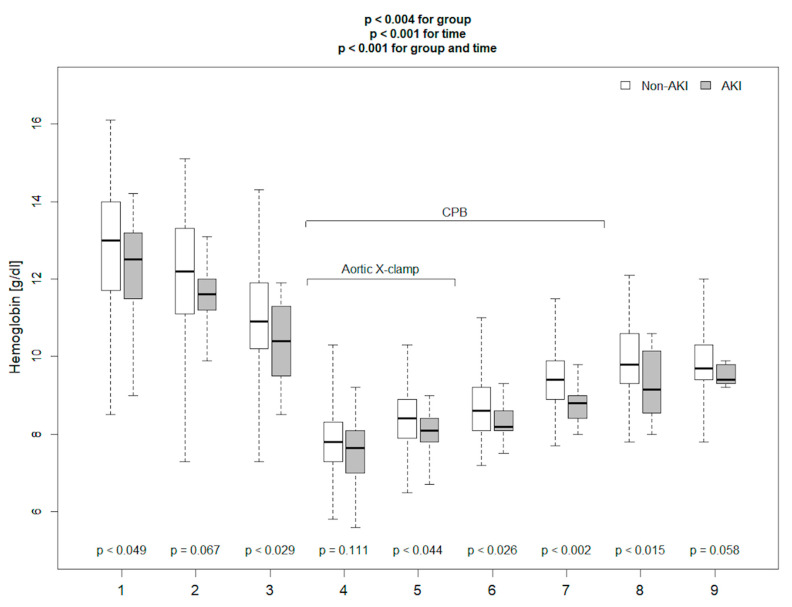
Haemoglobin concentration measured in the operating room (with *p* values for comparison between AKI and non-AKI patients at following time points). {1}—before anaesthesia induction, {2}—directly before skin incision, {3}—after sternum opening, {4}—20 min after aortic cross-clamping, {5}—40 min after aortic cross-clamping, {6}—20 min after aortic cross-clamp removal, {7}—20 min after weaning from CPB, {8}—40 min after separation from CPB, {9}—60 min after separation from CPB.

**Figure 2 diagnostics-11-01591-f002:**
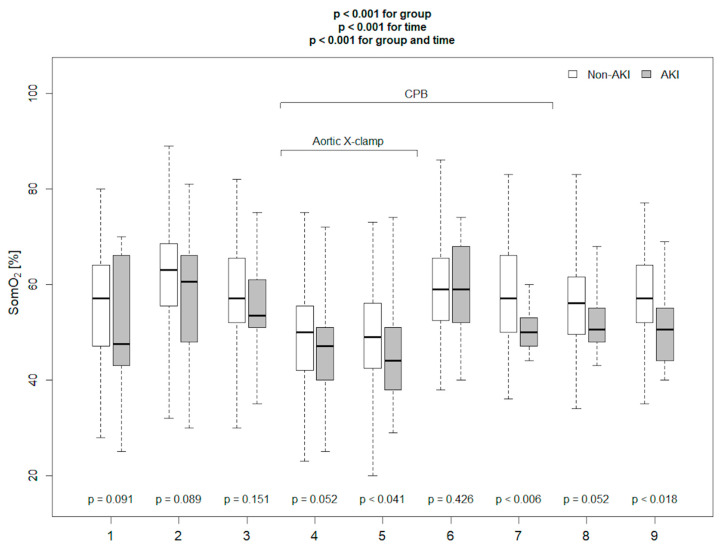
NIRS muscle saturation measured in the operation room in AKI and non-AKI patients at time-points specified in methods (*p* values for comparison between AKI and non-AKI patients at specified time points. {1}—before anaesthesia induction, {2}—directly before skin incision, {3}—after sternum opening, {4}—20 min after aortic cross-clamping, {5}—40 min after aortic cross-clamping, {6}—20 min after aortic cross-clamp removal, {7}—20 min after weaning from CPB, {8}—40 min after separation from CPB, {9}—60 min after separation from CPB).

**Figure 3 diagnostics-11-01591-f003:**
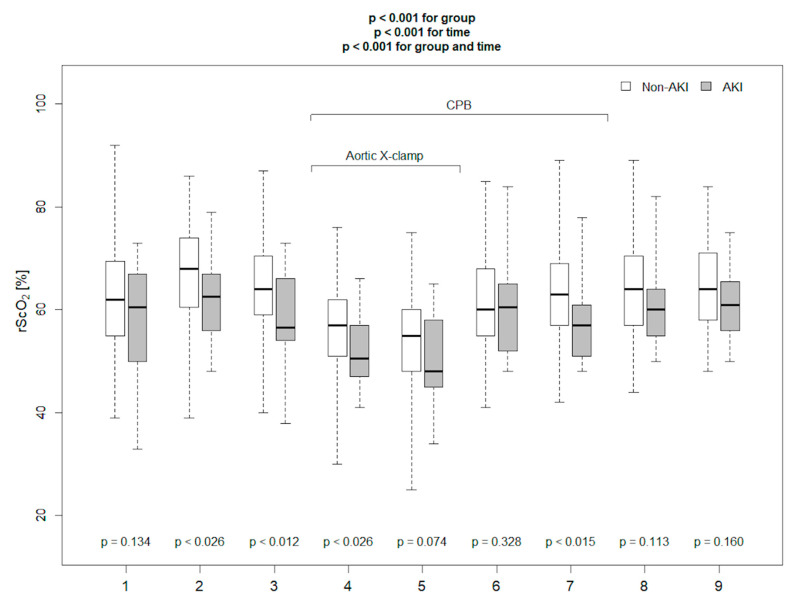
NIRS brain saturation measured in the operating room in AKI and non-AKI patients at time-points specified in methods (*p* values for comparison between AKI and non-AKI patients at specified time points. {1}—before anaesthesia induction, {2}—directly before skin incision, {3}—after sternum opening, {4}—20 min after aortic cross-clamping, {5}—40 min after aortic cross-clamping, {6}—20 min after aortic cross-clamp removal, {7}—20 min after weaning from CPB, {8}—40 min after separation from CPB, {9}—60 min after separation from CPB).

**Figure 4 diagnostics-11-01591-f004:**
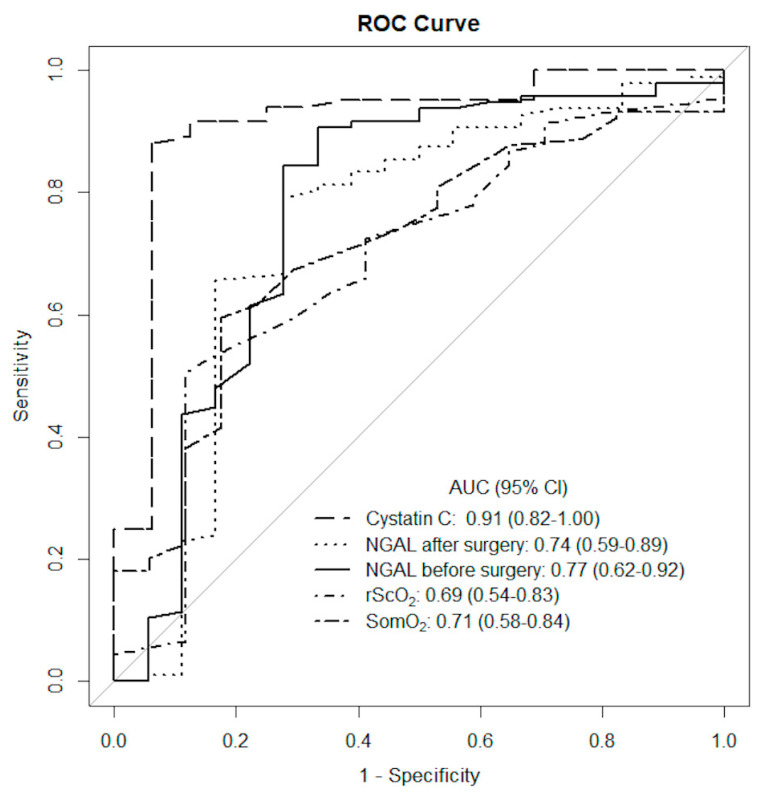
The accuracy of measured parameters as potential CS-AKI predictors based on the area (AUC) under the receiver-operating characteristic (ROC) curve.

**Table 1 diagnostics-11-01591-t001:** Demographic, clinical, and laboratory characteristics and data of cardiac surgery.

	AKI Patients *n* = 18	Non-AKI Patients *n* = 96	*p*
PREOPERATIVE CHARACTERISTICS			
Age (years)	74 (66–78)	67 (60–74)	0.029
Male, n (%)	8 (44)	49 (51)	0.798
Coronary artery disease, n (%)	12 (67)	62 (65)	1.000
Arterial hypertension, n (%)	12 (67)	74 (77)	0.376
Diabetes mellitus, n (%)	7 (39)	29 (30)	0.725
EUROScore (logistic)	9.7 (5.1–13.5)	5.5 (2.6–12.2)	0.054
LVEF%	60 (45–64)	60 (48–65)	0.440
Creatinine (mg/dL)	1.02 (0.91–1.27)	0.90 (0.79–1.05)	0.018
Haemoglobin (g/dL)	12.8 (11.8–13.3)	13.8 (12.6–14.6)	0.019
Preoperative anaemia *, n (%)	7 (39)	19 (20)	0.076
Leukocyte count (G/L)	6.83 (6.43–8.64)	7.34 (6.54–8.44)	0.166
Angiotensin-converting enzyme inhibitors/sartans before operation	4 (22)	28 (29)	0.776
Statins in premedication	10 (56)	63 (66)	0.433
INTRAOPERATIVE CHARACTERISTICS			
Aortic valve surgery, n (%)	6 (33)	61 (64)	0.021
Mitral valve surgery, n (%)	1 (5.6)	12 (12.5)	0.689
Aortic and mitral valves surgery, n (%)	3 (17)	4 (4)	0.077
Ascending aorta surgery including Bentall operation n (%)	3 (17)	6 (6)	0.150
3 valves surgery, n (%)	4 (22)	7 (7)	0.071
Other surgery, n (%)	1 (5.6)	6 (6)	1.000
CPB time (min)	140 (116–168)	119 (98–151)	0.084
Aortic cross-clamp time (min)	95 (83–112)	80 (67–103)	0.048
POSTOPERATIVE CHARACTERISTICS			
Serum creatinine on the 1st day post-surgery (mg/dL)	1.58 (1.31–1.95)	0.90 (0.79–1.05)	0.018
Serum creatinine on the 2nd day post-surgery (mg/dL)	1.52 (1.21–2.13)	0.87 (0.74–1.03)	0.001
Serum creatinine on the 3rd day post-surgery (mg/dL)	1.46 (0.99–1.67)	0.79 (0.68–0.91)	0.001
Catecholamine infusion on the 1st day post-surgery n (%)	10 (56)	29 (30)	0.059
Catecholamine infusion on the 2nd day post-surgery n (%)	7 (39)	12 (13)	0.014
Catecholamine infusion on the 3rd day post-surgery n (%)	4 (22)	9 (9)	0.223
Diuresis on the 1st day post-surgery (mL)	1940 (1400–2470)	2340 (2070–2690)	0.019
Fluid balance on the 1st day post-surgery (mL)	−20 (−769–918)	−650 (−923–−50)	0.042
Fluid balance on the 2nd day post-surgery (mL)	297 (−435–860)	0 (−950–400)	0.093
Fluid balance on the 3rd day post-surgery (mL)	−300 (−750–450)	−375 (−925–162.5)	0.288
Postoperative chest drainage on the 1st day post-surgery (mL)	435 (273–950)	320 (225–533)	0.083
Postoperative chest drainage on the 2nd day post-surgery (mL)	130 (120–235)	160 (98–255)	0.497
Furosemide on the 1st day post-surgery, n (%)	10 (56)	49 (51)	0.007
Time to extubation (hours)	10.0 (8.5–11.5)	8.0 (6.5–10.0)	0.028
CRP on the 1st day post-surgery	38.2 (25.0–54.7)	28.3 (17.5–42.7)	0.043
CRP on the 2nd day post-surgery	79.3 (60.3–108.6)	65.8 (44.0–93.82)	0.086
CRP on the 3rd day post-surgery	113.3 (62.0–153.8)	98.5 (62.3–142.6)	0.286
WBC on the 1st day post-surgery	12.5 (10.8–15.2)	12.9 (10.8–14.9)	0.342
WBC in the 2nd day post-surgery	15.0 (12.7–19.4)	14.0 (12.0–16.0)	0.117
WBC in the 3rd day post-surgery	11.5 (10.7–14.7)	10.3 (8.1–12.5)	0.037
Haemoglobin on the 1st day post-surgery	10.5 (9.7–11.3)	10.8 (10.1–11.4)	0.239
Haemoglobin on the 2nd day post-surgery	9.6 (9.1–10.7)	10.2 (9.6–10.7)	0.074
Haemoglobin on the 3rd day post-surgery	10.0 (8.9–10.5)	9.6 (8.9–10.4)	0.315
BIOMARKERS			
Blood NGAL before surgery (ng/mL)	123.5 (78.5–163.3)	62.5 (50.8–86.5)	0.001
Blood NGAL 3 h after surgery (ng/mL)	156.50 (94.00–181.00)	74.00 (53.75–101.25)	0.004
Postoperative cystatin C (mg/L)	1.56 (1.41–1.94)	0.84 (0.72–1.07)	0.001

Data are presented as median (25th–75th percentile) or numbers (and percent). Abbreviations: AKI—acute kidney injury; CPB—cardio-pulmonary bypass; CRP—C-reactive protein; LVEF—left ventricle ejection fraction; NGAL—neutrophil gelatinase-associated lipocalin; WBC—white blood cell. *—Preoperative anaemia was defined as haemoglobin level <13 g·dL^−1^ in men and <12 g·dL^−1^ in women.

**Table 2 diagnostics-11-01591-t002:** Cut-off values for pre- and postoperative blood NGAL, postoperative cystatin C, the absolute SomO_2,_ and the absolute rScO_2_ in CS-AKI prediction (based on AUC ROC analysis).

Parameters	Cut-off Value	AUC (95% CI)	Sensitivity	Specificity	PPV	NPV
Cystatin C after surgery * (mg/L)	1.23	91.4 (82.0–100.0)	0.88	0.94	0.99	0.6
Blood NGAL before surgery (ng/mL)	91.5	73.9 (58.5–89.3)	0.79	0.72	0.94	0.39
Blood NGAL 3 h after surgery (ng/mL)	140.5	77.1 (62.4–91.9)	0.91	0.67	0.94	0.57
The absolute SomO_2_ 20 min after CPB (%)	54.5	71.1 (58.1–84.0)	0.6	0.82	0.95	0.28
The absolute rScO_2_ 20 min after CPB (%)	62.5	68.6 (54.2–82.9)	0.51	0.88	0.96	0.25

Abbreviations: AUC—area under the receiver-operating characteristic (ROC) curve; CS-AKI—cardiac surgery-associated acute kidney injury; CPB—cardio-pulmonary bypass; NGAL—neutrophil gelatinase-associated lipocalin; NPV—negative predictive value; PPV—positive predictive value, rScO_2_—regional cerebral oxygen saturation measured by near-infrared spectroscopy, SomO_2_—somatic oxygen saturation of thenar muscles measured by near-infrared spectroscopy. * Postoperative cystatin C was obtained between 12 and 20 h after surgery.

**Table 3 diagnostics-11-01591-t003:** Univariate logistic regression analyses for blood NGAL, cystatin, and NIRS cut-off values in CS-AKI prediction.

Parameters	OR (95% CI)	*p*
NGAL before surgery ≥ 91.5 ng/mL	9.88 (3.15–30.98)	0.001
NGAL 3 h after surgery ≥ 140.5 ng/mL	19.33 (5.84–63.96)	0.001
Postoperative cystatin C ≥ 1.23 mg/L	111 (13.2–933.33)	0.001
SomO_2_ 20′ after CPB ≤ 54.5%	6.87 (1.3–13.97)	0.003
rScO_2_ 20′ after CPB ≤ 62.5%	3.5 (1.14–10.78)	0.003
Blood NGAL before surgery ≥ 91.5 nl/mL and SomO_2_ 20′ after CPB ≤ 54.5%	12.7 (3.88–41.59)	0.001
Blood NGAL before surgery ≥ 91.5 nl/mL and rScO_2_ 20′ after CPB ≤ 62.5%	17.45 (5.16–59.06)	0.001
Blood NGAL 3 h after surgery ≥ 140.5 ng/mL and SomO_2_ 20′ after CPB ≤ 54.5%	30.36 (7.54–122.16)	0.001
Blood NGAL 3 h after surgery ≥ 140.5 ng/mL and rScO_2_ 20′ after CPB ≤ 62.5%	39.88 (9.71–163.67)	0.001
Postoperative cystatin C ≥ 1.23 mg/L and SomO_2_ 20′ after CPB ≤ 54.5%	58.5 (12.32–276.84)	0.001
Postoperative cystatin C ≥ 1.23 mg/L and rScO_2_ 20′ after CPB ≤ 62.5%	123.5 (20.49–744.46)	0.001

Abbreviations: CS-AKI—cardiac surgery-associated acute kidney injury; CPB—cardio-pulmonary bypass; NGAL—neutrophil gelatinase-associated lipocalin; OR—Odds ratio; CI—confidence interval; rScO_2_—regional cerebral oxygen saturation measured by near-infrared spectroscopy, SomO_2_—somatic oxygen saturation of thenar muscles measured by near-infrared spectroscopy.

## Data Availability

Please refer to the corresponding author.

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
