# Peer review of "Brain and Muscle Oxygen Saturation Combined with Kidney Injury Biomarkers Predict Cardiac Surgery Related Acute Kidney Injury"

_diagnostics, 2021, doi:10.3390/diagnostics11091591_

Round 1
Reviewer 1 Report
I think the article is well written and may be published without changes. congratulations to the authorsAuthor Response
Response to the Reviewer 1
The authors wish to thank you for the professional review. We would also like to thank you for the valuable time you devoted to reviewing our article.
Reviewer 2 Report
The authors present a clear study of parameters that can be used to identify patients at risk for AKI following cardiac surgery and carry out clear statistical evaluation to support their analysis and conclusions.
One question concerns the observation that 60% of both the AKI and non-AKI patients had LVEF 45-65%, with the remaining 40% (30-45%) because of their exclusion of patients less than 30% LVEF...it would be of interest to comment on any potential differences observed between these groups
Author Response
Response to the Reviewer 2
The authors wish to thank for the professional review of our study. The Reviewer introduced essential new aspects and allowed the authors to approach their data with ample criticism. The authors tried to answer all suggestions and remarks with due attention and diligence to the limit of their study data. Additionally, according to the Reviewer’s suggestion, we used additional checking by a native English-speaking colleague.
Question 1
„The authors present a clear study of parameters that can be used to identify patients at risk for AKI following cardiac surgery and carry out clear statistical evaluation to support their analysis and conclusions. One question concerns the observation that 60% of both the AKI and non-AKI patients had LVEF 45-65%, with the remaining 40% (30-45%) because of their exclusion of patients less than 30% LVEF...it would be of interest to comment on any potential differences observed between these groups”
Answer 1.
The authors wish to thank for the professional and most valuable comments provided by the Reviewer. We performed this important comparison between the two groups and added the necessary data (Table S2, page 10, lines 230- 235) for further Reviewer’s acceptance.
Reviewer 3 Report
I read with interest the article from W Szymanowicz et al. In this work, the authors aimed at evaluate whether NIRS technology with evaluation of brain and muscle oxygen saturation, combined with known biomarker of acute kidney injury, could be valuable tool to better identify patients at risk of acute kidney injury in the setting of scheduled cardiac surgery.
In this single center-prospectiv observationnal study, 114 patients were included, in whom 18 experienced acute kidney injury, the authors state that NIRS evaluation could be a early parameter to the risk of AKI related to cardiac surgery.
The article is well written, despite few moderate english wording.
I have major concerns regarding this work.
- Near 10% of the patients were excluded due to lack of consent. How can this be explain in the setting of scheduled intervention? Were excluded patients different from the population at baseline? This bias must be discussed.
- I think that the choice in the outcome is debattable. The NGAL biomarker is known to be an interesting tool to detect renalinjury. In the cardiac surgery setting, do we really need tools to identify this phenomenon? Instead of this, a better prediction of acute kidney injury (renal dysfunction) is mandatory. So, would'nt it be interesting to focus on acute kidney injury occurence?
- Acute kidney injury is defined according to KDIGO classification. What was taken in account? Stage 1 defined acute kidney injury? Please precise it.
- There is no description of acute kidney injury events: AKI stage, AKI reached criteria (serum creatinin? diuresis? need for renal replacement therapy?), need for renal replacement therapy, modalities and duration of the support.
- As the biomarkers and NIRS try to detect the same event, a correlation study between this test should be performed.
- Based on small sample size, design of the study, and number of AKI, statistical power is really weak. All the conclusions must be tempered.
- For the definition of AKI with serum creatinin criteria, which creatinine measurement was taken in account? Pre operative characteristics out of emergency setting might be relevant for the evaluation of baseline creatinin, but this must be discussed and stress out, and if baseline creatinin in the 3 month preceding surgery is known, to be sure that no AKI was present before surgery.
- Baseline characteristics aret quite different between groups, that is interesting and must be discussed in the discussion: creatinine, anemia...
- I am very surprised that, in a population of patient undergoing cardiac surgery, with a known history of coronary artery disease or arterial hypertension in more than 2/3 of patient, only 22 and 29% received RAAS inhibitors. This must be discussed: was it stopped before the surgery? it is not detailed in the supplementary ressources.
- In predicting AKI occurence, tools with a sensitivity of 51 and 60% is not really a valuable tool, all the more if the authors looked for small changes in serum creatinine...
- It is very hard to know what to keep in mind with all the univariate logistic regression and after adjustment (Supplemental Table), with only 18 event...
In my opinion, the trial was interesting, but primary outcome is inappropriate, and the conclusion must be changed, I don't think that, with these results, NIRS could be evaluated as a valuable tool for post cardiac surgery prediction risk assessment.
Author Response
Response to the Reviewer 3
The authors wish to thank for the professional and most valuable comments provided by the Reviewer. The Reviewer introduced essential new aspects and allowed the authors to approach their data with ample criticism. The authors tried to answer all suggestions and remarks with due attention and diligence to the limit of their study data.
Question 1.
„Near 10% of the patients were excluded due to lack of consent. How can this be explain in the setting of scheduled intervention? Were excluded patients different from the population at baseline? This bias must be discussed.”
Answer 1.
The authors wish to thank the Reviewer for raising that important point here and tried to answer with due attention and diligence to the limit of their study data. Near 10% of the patients were excluded from the study due to lack of consent; therefore, that patients did not have biomarkers and NIRS parameters measurements, but the operations were performed, and the patients have been treated according to the clinical standards. Unfortunately, we did not collect the clinical data about that patients; therefore, it is difficult for us at that moment to compare these patients with the enrolled population. The question is clinically essential; therefore, according to the Reviewer’s suggestion, we added this information into the Limitations of the study for further Reviewer’s acceptance (page 12, lines 338 – 340).
Question 2.
„I think that the choice in the outcome is debatable. The NGAL biomarker is known to be an interesting tool to detect renal injury. In the cardiac surgery setting, do we really need tools to identify this phenomenon? Instead of this, a better prediction of acute kidney injury (renal dysfunction) is mandatory. So, wouldn't it be interesting to focus on acute kidney injury occurrence? „
Answer 2.
The authors wish to thank the Reviewer for raising that important point here. We completely agree that the choice of primary outcome is debatable. We changed it according to the Reviewer’s suggestion noting the CS-AKI as the primary outcome (page 2, line 70).
Question 3.
„Acute kidney injury is defined according to KDIGO classification. What was taken in account? Stage 1 defined acute kidney injury? Please precise it.”
Answer 3.
The authors wish to thank Reviewer for this valuable remark. We appreciate the reviewer’s insightful suggestion and add appropriate information in the text of Methods part (page 2, lines 72- 80) and now there is the mentioned below information: CS-AKI was defined as an increase in creatinine level by 0.3 mg/dL over 48 hours or 50% in 7 days; 3 stages of kidney damage were taken into account: stage 1 (1.5 to 1.9-fold increase in the initial creatinine concentration or increase above 0.3 mg/dL), stage 2 (2 to 2.9-fold increase in the initial creatinine concentration), and stage 3 (3-fold increase in baseline creatinine concentration or increase in creatinine concentration to the value ≥ 4 mg/dL or the need for renal replacement therapy).
Question 4.
There is no description of acute kidney injury events: AKI stage, AKI reached criteria (serum creatinine? diuresis? need for renal replacement therapy?), need for renal replacement therapy, modalities, and duration of the support.
Answer 4.
The authors wish to thank the Reviewer for this valuable remark. We appreciate the Reviewer’s insightful suggestion and add appropriate information in the Results’ part (page 3, lines 137 - 142). Now there is information that 18 patients (16%) were diagnosed with CS-AKI based on the KDIGO criteria, of which 12 were stage 1, in 4 - stage 2, and in 2 - stage 3 (in one of these patients, apart from the diagnostic increase in creatinine level oliguria was diagnosed in the postoperative period - this patient required renal replacement therapy in the postoperative period).
Question 5.
„As the biomarkers and NIRS try to detect the same event, a correlation study between this test should be performed”.
Answer 5.
The authors wish to thank the Reviewer for this valuable remark. We performed the necessary analysis. The correlations between biomarkers and NIRS parameters were weak; although some were statistically significant, the linear correlation coefficient r was below 0.5. We add this information in the Results’ part (page 10, lines 217- 219) and as supplementary material (Table S3) for further Reviewer’s acceptance.
Question 6.
„Based on small sample size, design of the study, and number of AKI, statistical power is really weak. All the conclusions must be tempered.”
Answer 6.
The authors wish to thank Reviewer for this valuable remark. We appreciate the Reviewer’s insightful suggestion and tried to do our best to tempered our conclusions (page 1, lines 30- 32 and page 13, lines 348 – 353). We changed the text for further Reviewer’s acceptance.
Question 7.
„For the definition of AKI with serum creatinine criteria, which creatinine measurement was taken in account? Pre-operative characteristics out of emergency setting might be relevant for the evaluation of baseline creatinine, but this must be discussed and stress out, and if baseline creatinine in the 3 month preceding surgery is known, to be sure that no AKI was present before surgery.”
Answer 7.
The authors wish to thank the Reviewer for this valuable remark. The creatinine measured the day before the operation was taken into account as the baseline. We appreciate the Reviewer’s insightful suggestion and added new information into the Methods section (page 2, lines 75- 76). The Reviewer’s suggestion that AKI might be present before surgery in some patients is supported by our finding that preoperative NGAL was also significant AKI predictor. Unfortunately, we did not collect information about creatinine three months preceding the surgery because patients were enrolled into the study one day before surgery and it was impossible to add this measurement into the study design. Due to the extensive clinical importance of this issue, we added this information in the Limitations of the study (page 12, lines 336 - 338).
Question 8.
„Baseline characteristics are quite different between groups, that is interesting and must be discussed in the discussion: creatinine, anemia...”
Answer 8.
The authors wish to thank the Reviewer for this valuable remark. We tried to extend our discussion according to the Reviewer’s suggestion for further Reviewer’s acceptance (page 11 and 12, lines 245 - 292, new positions 28 – 40 in the References).
Question 9.
„I am very surprised that, in a population of patient undergoing cardiac surgery, with a known history of coronary artery disease or arterial hypertension in more than 2/3 of patient, only 22 and 29% received RAAS inhibitors. This must be discussed: was it stopped before the surgery? it is not detailed in the supplementary resources.”
Answer 9.
The authors wish to thank Reviewer for this valuable remark. The Reviewer is entirely right that the percentage of RAAS inhibitors using was so low. We have to note that the RAAS inhibitors are not interrupted in our Department, and we continue the pre-hospital pharmacological treatment before the surgery. On the other hand, it is difficult to explain such results in our patients; the ultimate decision about pharmacological therapy before the operation in any patient was left to the discretion of the treating physician. Based on that reason, we could only add information about this discrepancy into the Limitations of the study (page 12, lines 341- 343).
Question 10.
„In predicting AKI occurrence, tools with a sensitivity of 51 and 60% is not really a valuable tool, all the more if the authors looked for small changes in serum creatinine...”
Answer 10.
The authors wish to thank Reviewer for this valuable remark. We fully agree with the Reviewer’s that the analyzed parameters (brain and muscle oxygenation) have low sensitivity and are not valuable indices as a single parameter. Therefore, we performed additional statistical analyses, where we tried to measure these parameters together with biomarkers (for which the sensitivity is really high). We tried to stress this really weak point of the study and now add this information in the Discussion part (page 12, lines 321- 325) for further Reviewer’s acceptance.
Question 11.
„It is very hard to know what to keep in mind with all the univariate logistic regression and after adjustment (Supplemental Table), with only 18 event...”
Answer 11.
The authors wish to thank Reviewer for this valuable remark. We precise this information in the Results part (page 10, lines 219 – 228) and add the information about supplementary Table S2 presenting these results for further Reviewer’s acceptance.
Round 2
Reviewer 3 Report
I would like to thank the authors for their work.
If the occurence of AKI is poorly determined with NIRS, is it the same whatever the AKI severity is?
Maybe it could be interesting to present NIRS findings according to the AKI severity? Maybe the worse the NIRS, the more severe the AKI?
All my other questions have been answered.
Author Response
Response to the Reviewer 3
The authors wish to thank for the professional and most valuable comments provided by the Reviewer. We would also like to thank you for the valuable time you devoted to reviewing our article. The Reviewer introduced further essential new aspect and allowed the authors to approach their data with ample criticism. The authors tried to answer the suggestion with due attention and diligence to the limit of their study data.
Question 1.
“ If the occurrence of AKI is poorly determined with NIRS, is it the same whatever the AKI severity is? Maybe it could be interesting to present NIRS findings according to the AKI severity? Maybe the worse the NIRS, the more severe the AKI?”
Answer 1.
The authors wish to thank the Reviewer for raising that important point here and tried to answer with due attention and diligence to the limit of their study data. The Reviewer is right that it is essential to check the usefulness of NIRS measurements according to the AKI severity. Unfortunately, based on the statistician's opinion, due to the small sample size and limited numbers of AKI patients in every stage, the statistical power of possible analyses would be weak. The Reviewer’s question is clinically essential; therefore, we raised that issue in the Discussion and Limitations of the study for further Reviewer’s acceptance (page 12, lines 324 – 327 and 331 – 334).
Sincerely yours,
Authors.